# Ultra-Marathon-Induced Increase in Serum Levels of Vitamin D Metabolites: A Double-Blind Randomized Controlled Trial

**DOI:** 10.3390/nu12123629

**Published:** 2020-11-25

**Authors:** Jan Mieszkowski, Błażej Stankiewicz, Andrzej Kochanowicz, Bartłomiej Niespodziński, Tomasz Kowalik, Michał A. Żmijewski, Konrad Kowalski, Rafał Rola, Tomasz Bieńkowski, Jędrzej Antosiewicz

**Affiliations:** 1Department of Gymnastics and Dance, Gdansk University of Physical Education and Sport, 80-336 Gdansk, Poland; mieszkowskijan@gmail.com (J.M.); andrzejkochanowicz@o2.pl (A.K.); 2Institute of Physical Education, Kazimierz Wielki University, 85-064 Bydgoszcz, Poland; blazej1975@interia.pl (B.S.); bartlomiej.niespodzinski@ukw.edu.pl (B.N.); tomasz.kowalik@ukw.edu.pl (T.K.); 3Department of Histology, Medical University of Gdańsk, 80-211 Gdansk, Poland; michal.zmijewski@gumed.edu.pl; 4Masdiag Sp. z o.o. Company, 01-882 Warsaw, Poland; konrad.kowalski@masdiag.pl (K.K.); r.rola@doktorant.umk.pl (R.R.); tomasz.bienkowski@masdiag.pl (T.B.); 5Chair of Environmental Chemistry and Bioanalytics, Faculty of Chemistry, Nicolaus Copernicus University in Toruń, 87-100 Toruń, Poland; 6Department of Bioenergetics and Physiology of Exercise, Medical University of Gdansk, 80-210 Gdansk, Poland

**Keywords:** endurance exercise, 3-*epi*-25(OH)D_3_, 24,25(OH)_2_D_3_, 25(OH)D_3_

## Abstract

Purpose: While an increasing number of studies demonstrate the importance of vitamin D for athletic performance, the effects of any type of exercise on vitamin D metabolism are poorly characterized. We aimed to identify the responses of some vitamin D metabolites to ultra-marathon runs. Methods: A repeated-measures design was implemented, in which 27 amateur runners were assigned into two groups: those who received a single dose of vitamin D_3_ (150,000 IU) 24 h before the start of the marathon (*n* = 13) and those (*n* = 14) who received a placebo. Blood samples were collected 24 h before, immediately after, and 24 h after the run. Results: In both groups of runners, serum 25(OH)D_3_, 24,25(OH)_2_D_3_, and 3-*epi*-25(OH)D_3_ levels significantly increased by 83%, 63%, and 182% after the ultra-marathon, respectively. The increase was most pronounced in the vitamin D group. Body mass and fat mass significantly decreased after the run in both groups. Conclusions: Ultra-marathon induces the mobilization of vitamin D into the blood. Furthermore, the 24,25(OH)_2_D_3_ and 3-*epi*-25(OH)D_3_ increases imply that the exercise stimulates vitamin D metabolism.

## 1. Introduction

Vitamin D plays a crucial role in the regulation of multiple physiological processes. Its activity is mainly ascribed to the active form, 1,25(OH)_2_D_3_, which acts via a specific vitamin D receptor (VDR). VDR is a transcriptional factor that regulates the expression of approximately 1000 genes. VDR is present in almost all human tissues [1,2]. Consistently, vitamin D deficiency has been associated with multiple morbidities, such as cancer, diabetes, multiple sclerosis, cardiovascular diseases, and others [3,4,5,6]. Therefore, it is recognized that vitamin D status is an important risk factor for several diseases of civilization. Moreover, more and more athletes also show a low vitamin D status, which may negatively impact the health, performance, and training efficiency of athletes [7,8].

Vitamin D is produced in the skin in response to ultraviolet (sunlight) exposure. Subsequently, it is hydroxylated at positions 25 and 1 to gain full hormonal activity. On the other hand, 25-OH vitamin D [25(OH)D_3_] is a good marker of vitamin D status. The kidney, brain, bone, skin, prostate, and white blood cells can convert 25(OH)D_3_ to its active form [1,25(OH)_2_D_3_]. It can be anticipated that low serum levels of 25(OH)D_3_ will limit the synthesis of the active form in all these tissues. Serum 25(OH)D_3_ levels are mainly determined by exposure to sunlight and vitamin D supplementation. In addition, higher fat tissue content is associated with lower serum 25(OH)D_3_ levels, possibly because of its ability to store vitamin D [9,10]. On the other hand, higher physical activity is associated with better vitamin D status, even though many athletes are vitamin D-deficient [11]. Vitamin D supplementation is the easiest way to correct its deficiency and single high doses have been demonstrated to be effective in short periods [12].

Among many factors, exercise-induced release of vitamin D from adipose tissue has been postulated as an important mechanism that leads to increased vitamin D levels in the blood [13]. However, the effect of exercise on vitamin D status is not fully understood. Of note, vitamin D metabolism is involved in the formation of other metabolites, such as 3-*epi*-25(OH)D_3_ and 24,25(OH)_2_D_3_. These metabolites are not considered to be physiologically active. However, based on recent studies, they play an important role in the regulation of general metabolism. For example, 3-*epi*-25(OH)D_3_ levels are associated with an improved cardiovascular risk profile, and 3-*epi*-1,25(OH)_2_D_3_, derived from 3-*epi*-25(OH)D_3_, effectively reduces blood parathormone without inducing changes in the plasma calcium levels [14,15]. In addition, 24,25(OH)_2_D_3_, considered to be an inactive form of vitamin D, protects cells from 1,25(OH)_2_D_3_ toxicity and modulates the antioxidant potential by binding catalase [16]. Furthermore, studies involving animal models demonstrated that 24,25(OH)_2_D_3_ plays an important role in normal bone integrity, function, and healing [17,18]. Interestingly, high 24,25(OH)_2_D_3_ levels are associated with a reduced disability status in multiple sclerosis patients [19].

Collectively, the above observations indicate that vitamin D metabolites have important biological functions, which are far from being completely understood. Therefore, it is crucial to study the effects of exercise on vitamin D metabolism. Exercise stimulates the release of several hundreds of proteins (myokines) into circulation from the skeletal muscle while also stimulating the liberation of bioactive proteins (exerkines) from other tissues [20]. An example of such an exerkine is fibroblast growth factor 23, whose concentration increases after exercise [21]. This exerkine is responsible for the regulation of plasma phosphate levels and modifies vitamin D metabolism by inhibiting the formation of 1,25(OH)_2_D_3_ [22].

Ultra-marathon is a type of exercise that pushes the boundaries of human performance. This extreme type of physical activity, involving continuous running over a distance well above the 42 km of a regular marathon run, is associated with enormous energy expenditure. Until now, knowledge about vitamin D metabolism associated with strenuous exercise like ultra-marathon has been limited [23]. Running and walking for extreme durations, so-called ultra-marathons, have become increasingly popular in the last years throughout the world, particularly in the USA, Europe, Japan, and South Africa [24]. Hence, it seems important to evaluate the effect of this type of prolonged exercise on the physiological responses of the human body, both under conditions of supplementation and where supplementation is not provided. Especially vitamin D supplementation could prevent exercise-induced inflammation processes and other adverse body reactions. We proposed that ultra-marathon, which alters the production of hundreds of exerkines [25] and has the potential to reduce the amount of adipose tissue [26], influences vitamin D metabolism.

Here, we performed a double-blind randomized controlled trial to determine the impact of extreme endurance exercise on vitamin D metabolites in relation to vitamin D supplementation. We found that ultra-marathon induced a significant increase in metabolites of vitamin D_3_ which do not possess classical metabolic effects of the active form of vitamin D_3_.

## 2. Materials and Methods

### 2.1. Experimental Overview

The study was designed as a double-blind randomized controlled trial with parallel groups. Participants were randomly assigned to two groups: the supplementation group and the control group. The supplementation protocol involved a single high dose of vitamin D3 before the start of the ultra-marathon. During the initial visit, data on the subject’s age, body composition, and height were collected. All runners were examined by a professional physician. A sample of venous blood was obtained before the ultra-marathon start and immediately after and 24 h after the run to evaluate the vitamin D metabolites. Additionally, before starting the actual experiment, to evaluate body responses to a high dose of vitamin D, profiles of vitamin D analogues were assessed. All laboratory analyses were performed at the Gdansk University of Physical Education (Gdansk, Poland).

### 2.2. Participants

Twenty-seven male ultra-marathon runners taking part in the Lower Silesian Mountain Runs Festival Ultra-Marathon Race participated in the study. The runners were randomly assigned to the experimental (supplemented, UM-S; *n* = 13) and control (placebo, UM-C; *n* = 14) groups. The characteristics of the groups are shown in Table 1. The participants were physically active ultra-marathon amateur runners. None of the runners had any history of known diseases or reported any intake of medication due to illnesses in six months before the experiment. All runners had previous ultra-marathon race experience (not less than two). During all testing periods and 1 week before testing, the participants refrained from alcohol, caffeine, guarana, theine, tea, chocolate, and any other substance intake which may potentially influence exercise performance. Furthermore, the participants were asked to adopt a similar eating pattern on the days of measurements, based on a randomized diet for their age group and physical intensity. The study protocol was accepted by the Bioethics Committee for Clinical Research at of the Collegium Medicum University of Nicolaus Copernicus (decision number KB-124/2017) and conducted according to the Declaration of Helsinki. The study was registered as clinical trial: NCT03417700. Written informed consent was obtained from all study participants, who were also informed about the possibility of withdrawal of consent at any time and for any reason. Prior to participation, subjects were informed about the study procedures but not about the rationale and study aim, so as to keep them naive about the potential effect of supplementation.

### 2.3. Pilot Study

Although vitamin D metabolism is well documented, to date, changes in serum levels of vitamin D metabolites after administration of a high dose of vitamin D have not been evaluated. To evaluate body responses to a high dose of vitamin D, profiles of vitamin D analogues were assessed. For this purpose, four physically active non-ultra-marathon runners (volunteers) took two doses of vitamin D3 (100,000 or 200,000 IU) 28 day apart. The blood samples were taken at selected time points (days 0 to 45) and profiles of vitamin D analogues were analyzed as described in Section 2.6. The following were analyzed: 25(OH)D_3_, 3-*epi*-25(OH)D_3_, and 24,25(OH)_2_D_3_, and the ratios of the last two compounds to 25(OH)D_3_.

### 2.4. Ultra-Marathon Run

One day after the first blood sample collection, physician examination, and supplementation, all runners (UM-C and UM-S groups) participated in the Lower Silesian Mountain Run Festival (19 July 2018). The start and finish points were in the town of Lądek Zdrój (Lower Silesian Voivodeship, Poland).

The running festival took place in the Kłodzko Land (latitude of 50° N) and consisted of seven mountain trails, with a maximum course length of 240 km, a maximum altitude of approximately 1425 m a.s.l., and a minimum altitude of approximately 261 m a.s.l. The entire altitude range was approximately 1164 m, and the total ascent and descent was 7670 m. (Figure 1). The run started at 18:00 h and the temperature during the run varied from 18 °C at the start point to 4 °C on the top of the Śnieżnik Mountain. Most of the time, the sky was overcast.

### 2.5. Vitamin D Supplementation

Each participant from the experimental group received a single high dose of vitamin D3 (150,000 IU) 24 h before the start of the ultra-marathon. The decision was based on the pilot study, which showed the highest concentration of vitamin D metabolites within 24–48 h. The control group received a placebo solution with the taste (anise), color, and consistency matching those of the vitamin D solution (pure vegetable oil solution).

The participants and researchers had no knowledge of the groups and differences in the supplementation procedures.

### 2.6. Sample Collection and Measurements of Vitamin D Metabolite Levels

Blood (9 mL) was collected three times: 24 h before and after the race and immediately after the run (up to 5 min after the run). Venous blood samples were collected into Sarstedt S-Monovette tubes (S-Monovette^®^ Sarstedt AG&Co, Nümbrecht, Germany) without anticoagulant for serum separation (with a coagulation accelerator). The serum was separated using standard laboratory procedures, aliquoted, and frozen at −80 °C until further analysis. Sample preparation was based on serum protein precipitation and derivatization. Quantitative analysis was performed using liquid chromatography coupled with tandem mass spectrometry (QTRAP^®^4500 (Sciex) coupled with ExionLC HPLC system) with minor changes according to previously published method [27]. Serum samples were analyzed in the positive ion mode, using electrospray ionization. The raw data were collected using Analyst^®^ software, while to process and quantify the collected data MultiQuant^®^ was used. Various reagents were used in the sample preparation procedure. Furthermore, 4-(4′-Dimethylaminophenyl)-1,2,4-triazoline-3,5-dione (DAPTAD) was used as a derivatization agent. It was synthesized by Masdiag Laboratory (Warsaw, Poland). Additionally, solvents such as water, ethyl acetate (POCh S.A., Gliwice, Poland), and methanol (Honeywell, Sigma-Aldrich, Gillingham, Dorset, UK) were used.

Mobile phases were prepared using acetonitrile (ACN) (Honeywell, Sigma-Aldrich, Gillingham, Dorset, UK), water (POCh S.A., Gliwice, Poland), and formic acid (FA) (Merck KGaA, Darmstadt, Germany). All solvents were of LC-MS grade.

The following were determined: 25(OH)D_3_, 24,25(OH)_2_D_3_, 3-*epi*-25(OH)D_3_, and 25(OH)D_2_ levels and the ratios of 25(OH)D_3_ to 24,25(OH)_2_D_3_ and 25(OH)D_3_ to 3-*epi*-25(OH)D_3_. The concentrations of vitamin D metabolites were corrected to change in plasma volume [28,29].

### 2.7. Statistical Analysis

Descriptive statistics included mean ± standard deviation (SD) for all measured variables. A two-way ANOVA with repeated measures (2 × 3) was performed to investigate the impact of ultra-marathon running (marathon: 24 h before, immediately after, and 24 h after the run) on vitamin D metabolites and physical characteristics in relation to vitamin D supplementation (group: UM-S, UM-C). In case of a significant interaction, Tukey’s post-hoc test was performed to assess differences in specific subgroups. In addition, the effect size was determined by eta-squared statistics (η^2^). Values equal to or more than 0.01, 0.06, and 0.14 indicated a small, moderate, and large effect, respectively. All calculations and graphics were conducted using Statistica 12 software (StatSoft, Tulsa, OK, USA). Differences were considered statistically significant when *p* ≤ 0.05.

## 3. Results

In the control study, four volunteers who were not participating in any sport activities were given a single high dose of vitamin D on days 0 and 28, with two doses tested. Regardless of the dose, the 25(OH)D_3_ and 3-*epi*-25(OH)D_3_ levels increased within the first 48–72 h, following which the concentration gradually decreased. A similar relationship was observed for the 24,25(OH)_2_D_3_ metabolite, with the highest concentration noted between days 4 and 7 after administration. Its levels remained relatively constant once the peak was reached. Interestingly, the (slow) increase in the 25(OH)D_3_ levels began 12 h after vitamin D administration, while the 24,25(OH)_2_D_3_ levels started to increase 24 h after vitamin D administration (Figure 2).

Changes in the serum levels of vitamin D after the ultra-marathon are presented in Figure 3. The two-way ANOVA revealed a significant ultra-marathon effect in all analyzed variables on serum of vitamin D_3_ levels (Table 2). Regardless of vitamin D_3_ supplementation, a significant increase in the levels of 25(OH)D_3_ (82.9%), 24.25(OH)_2_D_3_ (63.3%), and 3-*epi*-25(OH)D_3_ (182.6%), 25(OH)D_2_ (17.7%) and the ratio of 25(OH)D_3_ to 24,25(OH)_2_D_3_ (45.8%) was observed, as well as a significant decrease in the ratio of 25(OH)D_3_ to 3-*epi*-25(OH)D_3_ (18.4%) immediately after the ultra-marathon. A significant increase in the levels of 24,25(OH)_2_D_3_, 3-*epi*-25(OH)D_3_, and 25(OH)D_2_ and the ratios of 25(OH)D_3_ to 24,25 (OH)_2_D_3_ and 25(OH)D_3_ to 3-*epi*-25(OH)D_3_ was also apparent 24 h after the ultra-marathon (Table 2). The two-way ANOVA also revealed a significant group effect in 25(OH)D_3,_ 3-*epi*-25(OH)D_3_, and 25(OH)D_3_: 24,25(OH)_2_D_3_ ratio (Table 2). The UM-S group showed a 30.1% higher concentration of 25(OH)D_3_, a 61.8% higher concentration of 3-*epi*-25(OH)D_3_, and a 24.7% higher ratio of 25(OH)D_3_ to 24,25(OH)_2_D_3_ in comparison with the UM-C group.

Furthermore, the analysis of variance of the 25(OH)D_3_ and 3-*epi*-25(OH)D_3_ levels and the ratio of 25(OH)D_3_ to 3-*epi*-25(OH)D_3_ also showed a significant interaction of the group and ultra-marathon factor. An interaction analysis of the ‘group’ and ‘ultra-marathon’ factors indicated that the 25(OH)D_3_ and 3-*epi*-25(OH)D_3_ levels and the ratio of 25(OH)D_3_ to 24,25(OH)_2_D_3_ immediately after and 24 h after the ultra-marathon were significantly higher in the group supplemented with vitamin D_3_ than in the control group (Figure 3).

## 4. Discussion

The main goal of the current study was to define whether vitamin D supplementation (150,000 IU) affects vitamin D metabolism after an acute exercise, such as an ultra-marathon. We demonstrated that serum 25(OH)D_3_, 24,25(OH)_2_D_3_, and 3-*epi*-25(OH)D_3_ levels significantly increased after the ultra-marathon. We also showed that in volunteers supplemented with large dose of vitamin D, similar changes were observed after the run; however, the time of changes differed. In addition, the control experiment demonstrated that application of single high dose of vitamin D is effective in correcting vitamin D deficiency, as reported before [12]. This observation indicates that exercise in addition to supplementation modifies vitamin D metabolism.

According to several reports, vitamin D is stored in adipose tissue and, therefore, increased lipolysis observed during exercise, which leads to vitamin D release into the blood [30,31]. In fact, 30 min of cycling increases 25(OH)D_3_ levels by approximately 20 nmol/L, while 5 weeks of progressive endurance exercise increases 25(OH)D levels by 2 nmol/L [13,32]. We confirmed, here, these observations, as we noted almost a 20 nmol/L increase in 25(OH)D levels in the control group runners and an even higher increase in the vitamin D-supplemented runners. However, contrary to the findings of Sun et al. [32], in the current study, 25(OH)D levels did not return to baseline values after 24 h but remained elevated. Detailed analysis of individual responses revealed that in 3 of 14 runners from the control group, 25(OH)D_3_ levels did not increase and even slightly decreased. On the other hand, 25(OH)D_3_ levels did not change after the run in two runners from the vitamin D-supplemented group. This could be explained by recent studies concerning the responders and non-responders to vitamin D supplementation; however, the detailed molecular mechanism of such variable responsiveness remains to be determined [33]. It cannot be excluded that vitamin D released from adipose tissue is reabsorbed by adipose tissue, and the reabsorption is affected by exercise to a lesser degree than the release. This would explain an earlier observation of a rapid return of 25(OH)D_3_ levels after exercise to the initial value [13]. The upper body subcutaneous adipose depot is a more important source of plasma fatty acids during exercise than visceral adipose tissue [34]. In addition, visceral adipose tissue accumulates more vitamin D than subcutaneous adipose tissue [35]. Hence, subtle differences in visceral adipose tissue may significantly affect serum levels of vitamin D and vitamin D metabolism during and after exercise. In addition, the adipose tissue content of vitamin D may be significantly different between individuals (4 to 500 ng/g) [36], which could also partially explain the different responses to the ultra-marathon.

In one study involving team-sport athletes, 12-week vitamin D supplementation resulted in increased serum 25(OH)D_3_ levels; however, a significant increase in 24,25(OH)_2_D_3_ levels was observed only after 70,000 IU of vitamin D_3_ was administered per week, while half of this dose had no effect [37]. Certainly, the observed effects were related to both, supplementation and training. Here, we observed a significant increase in 24,25(OH)_2_D_3_ levels in both supplemented and non-supplemented runners. This indicates an increased hydroxylation of vitamin D in C-24 position as a result of extreme exercise, such as an ultra-marathon. In addition, no direct correlation between an increase in 25(OH)D_3_ and 24,25(OH)_2_D_3_ levels was apparent in the above study [37]. That is probably associated with the activation of 24-hydroxylase after a certain cellular 25(OH)D_3_ threshold is exceeded. In the current study, 25(OH)D_3_ levels after the ultra-marathon were approximately 40 and 35 ng/mL in runners with and without supplementation, respectively. These levels were much lower than those observed in athletes supplemented with a lower dose of vitamin D (approximately 60 ng/mL) in whom changes in 24,25(OH)_2_D_3_ levels were not observed [37]. Furthermore, the ratio of 25(OH)D_3_ to 24,25(OH)_2_D_3_ also significantly increased after the ultra-marathon, which indicated a lack of direct dependency between these two compounds, implying that the exercise stimulates the synthesis of 24,25(OH)_2_D_3_. Conversely, in the control subjects, the ratio of 25(OH)D_3_ to 24,25(OH)_2_D_3_ initially rapidly increased as a result of a faster increase in the 25(OH)D_3_ levels than that of the 24,25(OH)_2_D_3_ levels. Over time, the ratio decreased, which was associated with a further production of 24,25(OH)_2_D_3_.

Exercise induces the release of several myokines and exerkines [20,38], and some of these molecules possibly stimulate C-24 hydroxylation. However, this requires further study. The 24,25(OH)_2_D_3_ metabolite is considered to be an inactive form of vitamin D. Nevertheless, according to recent studies, this metabolite has many biological functions, including protection against 1,25(OH)_2_D_3_ toxicity, reduction in inflammation, stimulation of bone healing, and some others [17,39]. Hence, an increase in its levels during exercise could have important implications for the body that should be investigated in future studies.

Another metabolite, whose concentration increased after the ultra-marathon, was 3-*epi*-25(OH)D_3_. C-3 epimerization is a common metabolic pathway of major metabolites of vitamin D_3_. 25(OH)D_3_ undergoes epimerization and 3-*epi*-25(OH)D_3_ is the most prevalent form [40]. The biological function of 3-*epi*-25-(OH)D_3_ is not well understood. Its concentration has been reported to be in the range of 0–9.01 ng/mL [41]. In the current study, 3-*epi*-25-(OH)D_3_ was detectable before and after the ultra-marathon in all runners. Interestingly, 3-*epi*-25(OH)D_3_ can be converted to 3-*epi*-1,25-(OH)D_3_, which participates in the suppression of parathormone secretion without inducing hypercalcemia and induces surfactant phospholipid synthesis in pulmonary cells [15,42]. To the best of our knowledge, this is the first paper reporting an exercise-induced increase in 3-*epi*-25(OH)D_3_ levels. The physiological meaning of this changes remains to be determined. Further, the relative contribution of 3-*epi*-25(OH)D_3_ to serum vitamin D does not correlate with 25(OH)D_3_ levels in individuals with hypervitaminosis D [43]. Conversely, here, we showed a decrease in the ratio of 25(OH)D_3_ to 3-*epi*-25(OH)D_3_ after the run or supplementation, indicating that increased serum 25(OH)D_3_ levels lead to a rise in C-3 epimerization. Hence, epimerization may be the first line of defense of the body against high levels of 25(OH)D_3_, since epimeric forms of 1,25(OH)_2_D_3_ are considered to be less biologically active than native forms.

The consumption of vitamins during an ultra-marathon is a common nutritional habit and, along with the growing interest in this type of physical activity, has been studied [44,45]. It should be highlighted that intake of typical antioxidants, such as vitamins C and E, as anti-inflammatory and antioxidative factors in endurance training could even blunt training adaptations and attenuate some of the cellular adaptations in skeletal muscle [46]. Furthermore consumption of those vitamins did not affect physiological aspects related to sport performance and did not improve sport results [47]. That is why there is a need for searching for proper supplementation methods, where vitamin D supplementation can be beneficial for ultra-marathon runners’ health.

In conclusion, this is the first report demonstrating that endurance exercise significantly increases serum levels of 24,25(OH)_2_D_3_, 3-*epi*-25(OH)D_3_, and 25(OH)D_3_, possibly by liberating vitamin D from adipose tissue and stimulating its metabolism. These observations imply that formation of vitamin D metabolites can, on the one hand, protect from vitamin D toxicity, and on the other hand, they can exert some other biological functions, e.g., anti- inflammatory and antioxidative. All of these data indicate that these changes in vitamin D metabolism are a physiological response to endurance exercise. The changes are affected by vitamin D status; thus, one can predict that low adipose or skeletal muscle vitamin stores may negatively influence physiological response to exercise. However, more work is needed to explore the role of vitamin D metabolites in physiological response to exercise.

## Figures and Tables

**Figure 1 nutrients-12-03629-f001:**
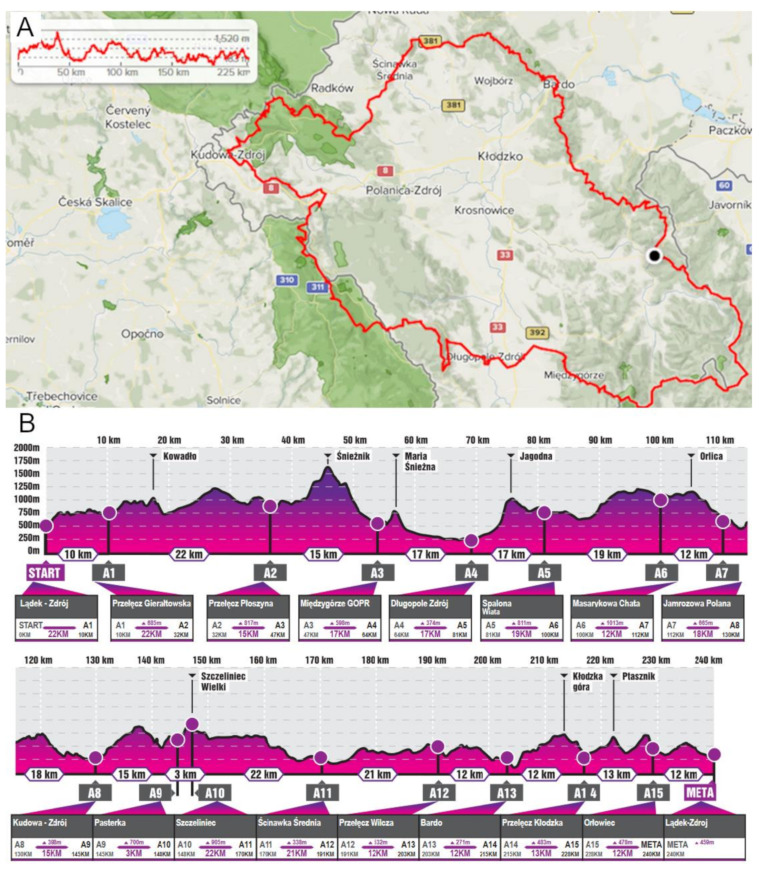
Ultra-marathon track characteristics of the Lower Silesian Mountain Run Festival 2018, Lądek Zdrój. The entire track (**A**) and select track parts and distances (**B**) are shown (Mountain Marathons Foundation).

**Figure 2 nutrients-12-03629-f002:**
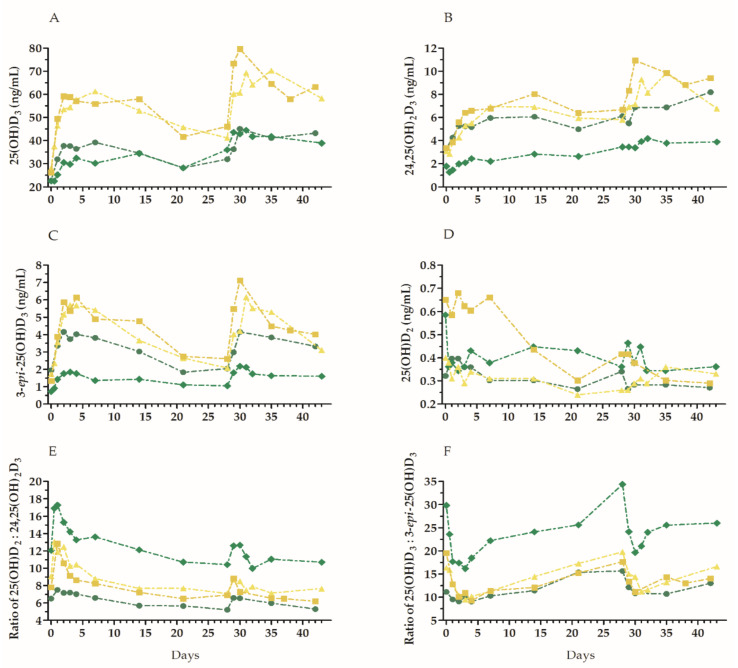
Changes in vitamin D metabolite levels in the serum, and metabolite ratios over time, in four healthy volunteers after receiving two doses of vitamin D_3_. The doses were given on days 0 and 28. Notation: green lines and symbols—participants administered with 100,000 IU; yellow lines and symbols—participants administered with 200,000 IU. The following were determined: 25(OH)D_3_ levels (**A**); 24,25(OH)_2_D_3_ levels (**B**); f25(OH)D_3_ levels (**C**); 25(OH)D_2_ (**D**); ratio of 25(OH)D_3_ to 24,25(OH)_2_D_3_ (**E**); ratio of 25(OH)D_3_ to 3-*epi*-25(OH)D_3_ (**F**).

**Figure 3 nutrients-12-03629-f003:**
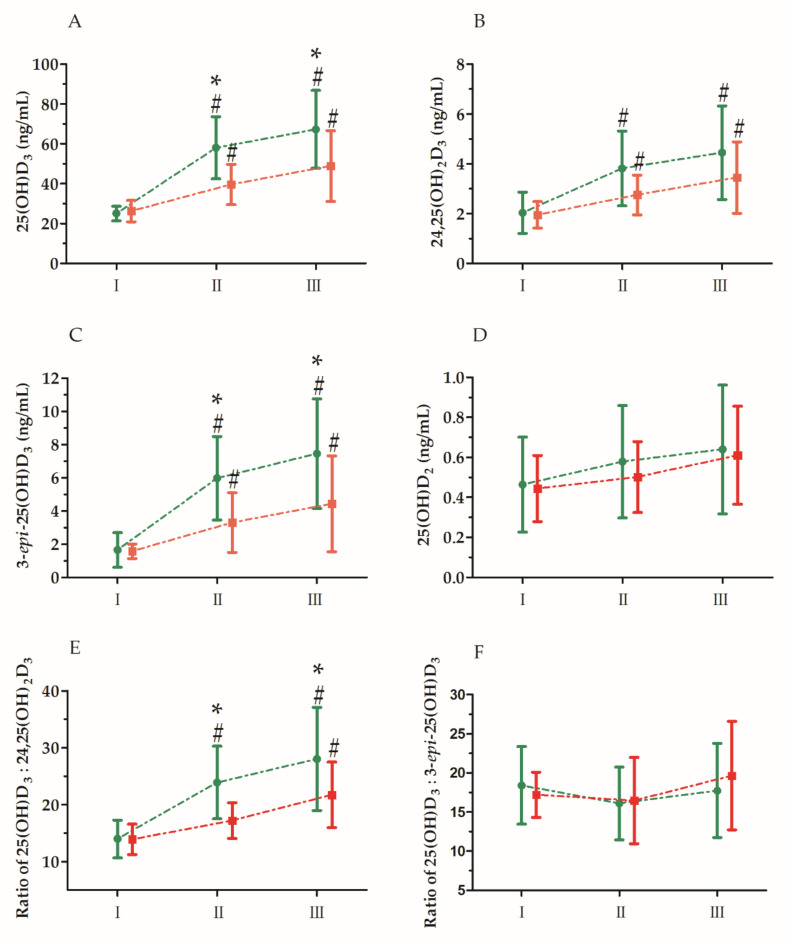
An effect of ultra-marathon on the serum levels of vitamin D metabolites. Data for runners supplemented with vitamin D_3_ (green symbols and lines) and runners without supplementation (red symbols and lines) are shown. (**A**) 25(OH)D_3_ levels; (**B**) 24,25(OH)_2_D_3_ levels; (**C**) 3-*epi*-25(OH)D_3_ levels; (**D**) 25(OH)D_2_ levels; (**E**) 25(OH)D_3_: 24,25(OH)_2_D_3_ ratio; (**F**) 25(OH)D_3_: 3-*epi*-25(OH)D_3_ ratio. Time points: I, 24 h before the ultra-marathon; II, immediately after the ultra-marathon; III, 24 h after the ultra-marathon. The values are presented as mean ± SD. * significant difference vs. group without supplementation at *p* < 0.05; ^#^ significant difference vs. I time point at *p* < 0.05.

**Table 1 nutrients-12-03629-t001:** Characteristics of the participants (*n* = 27).

Variable	UM-S (*n* = 13)		UM-C (*n* = 14)		Effect Size (η^2^)
	Mean ± SD	(95% CI)	Mean ± SD	(95% CI)	
Age (years)	42.00 ± 8.44	(36.00–47.00)	40.00 ± 8.11	(36.00–45.00)	0.01
Body mass	74.29 ± 7.51	(70.12–78.45)	78.64 ± 10.66	(72.20–85.09)	0.05
Body height (cm)	174.80 ± 3.80 *	(172.54–177.45)	181.30 ± 5.43	(178.43–183.56)	0.34
Body mass index	23.92 ± 2.42	(21.10–25.65)	24.18 ± 1.83	(22.95–25.41)	0.01
Fat mass (%)	12.58 ± 3.25	10.25–14.90	12.43 ± 4.65	9.31–15.56	0.02

Note: UM-S, runners given vitamin D3; UM-C, runners without vitamin D supplementation (control group). Significant difference at * *p* ≤ 0.05.

**Table 2 nutrients-12-03629-t002:** Two-way (2 groups × 3 repeated measurements) ANOVA of the serum levels of vitamin D_3_ induced by ultra-marathon run.

Variable	Effect	F	df	*p*	Effect Size (η^2^)	Post-Hoc Outcome
25(OH)D_3_	GR UM GR × UM	6.59 67.00 7.43	1, 30 2, 60 2, 60	0.01 * 0.01 ** 0.01 **	0.19 0.70 0.20	S > C I < II, III S-I < S-II, S-III C-I < C-II, C-III S-II > C-II S-III > C-III
24,25(OH)_2_D_3_	GR UM GR × UM	2.91 49.90 3.72	1, 30 2, 60 2, 60	0.09 0.01 ** 0.02 *	0.08 0.62 0.11	I < II < III S-I < S-II, S-III C-I < C-II, C-III
3-*epi*-25(OH)D_3_	GR UM GR × UM	7.84 58.32 7.66	1, 30 2, 60 2, 60	0.01 ** 0.01 ** 0.01 **	0.21 0.66 0.20	S > C I < II < III S-I < S-II, S-III C-I < C-II, C-III S-II > C-II S-III > C-III
25(OH)D_2_	GR UM GR × UM	0.26 34.06 1.05	1, 30 2, 60 2, 60	0.61 0.01 ** 0.35	0.01 0.53 0.03	I < II < III
Ratio 25(OH)D_3_: 24,25(OH)_2_D_3_	GR UM GR × UM	6.79 70.82 8.06	1, 30 2, 60 2, 60	0.01 * 0.01 ** 0.01 **	0.18 0.70 0.21	S > C I < II < III S-I < S-II, S-III C-I, C-II < C-III S-II > C-II S-III > C-III
Ratio 25(OH)D_3_: 3-*epi*-25(OH)D_3_	GR UM GR × UM	0.06 47.38 1.34	1, 30 2, 60 2, 60	0.81 0.04 * 0.26	0.01 0.10 0.04	II < III

Note: GR, group; UM, ultra-marathon; S, runners supplemented with vitamin D_3_; C, runners without supplementation (control group); I—24 h before the ultra-marathon; II—immediately after the ultra-marathon; III—24 h after the ultra-marathon. Significant difference at * *p* ≤ 0.05, ** *p* ≤ 0.01.

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
