# Peer review of "Ultra-Marathon-Induced Increase in Serum Levels of Vitamin D Metabolites: A Double-Blind Randomized Controlled Trial"

_nutrients, 2020, doi:10.3390/nu12123629_

Round 1

Reviewer 1 Report

The authors conducted a double-blind study in ultra marathon race to reveal the effect of ultra endurance exercise on the metabolism of vitamin D3. The experiments were carefully prepared and the measurement of vitamin D derivatives was done in an appropriate manner. The results obtained are novel and reliable. 

Major points

Table 1: There are two areas where the interactions do not appear to be significantly different: 25(OH)D2 and 25(OH)D3: epi-25(OH)D3

Minor points

Line 23: one dose of vitamin D"3".

Line 113: Indicate your weight, please.

Line 133: Please indicate the sunshine conditions on race day

Line 145: “ after "intake." should be removed.

Line 157: "SHIMADZU" LC-MS 8050.

Fig. 2: There are drastic increases 30-days after supplementation and these peaks seems artificial. The reviewer recommends to plot from 0 to 30 days.

Fig. 3: There are three graphs where the "2" on the y-axis of the graphs are not subscripted: 24,25(OH)2D3, 25(OH)D"2", and ratio 25(OH)D3:24,25(OH)2D3.

Author Response

The authors would like to thank the editor and reviewers for the opportunity to revise and for all comments that will improve the quality of the manuscript. Please find below a direct response to the comments. Please notice that all line indications refer to the version with highlighted changes. Aside from the comments below, to clarify all the methods, we introduced the extended information about procedures in lines: 174-176, 177-185, and 187-188.

R1

The authors conducted a double-blind study in ultra marathon race to reveal the effect of ultra endurance exercise on the metabolism of vitamin D3. The experiments were carefully prepared and the measurement of vitamin D derivatives was done in an appropriate manner. The results obtained are novel and reliable.

Major points

Table 1: There are two areas where the interactions do not appear to be significantly different: 25(OH)D2 and 25(OH)D3: epi-25(OH)D3

Answer: thank you for this comment. We assumed that the Reviewer refers to figure 3 as table 1 shows the participant's characteristics. Indeed the analysis did not show the significance of the interaction for the 25(OH)D2 and 25(OH)D3: epi-25(OH)D3. Considering 25(OH)D2, it was recently (Martineau, Thummel et al. 2019)showed that even 4 months of Vitamin D3 (2.5 mg) supplementation was unable to increase significantly the level of 25(OH)D2. What is more, the concentration of this vitamin D metabolite is much lower and more variable (Freeman, Wilson et al. 2015)in comparison to D3 fractions, thus any changes are more difficult to detect.

Despite fact that both 25(OH)D3 and epi-25(OH)D3 increased their concentrations more in the supplemented group, the relation of 25(OH)D3 to epi-25(OH)D3 did not show a difference between groups. As the investigated ratio is derivative of the other two vitamin D metabolites, the reason for that is mostly due to a similar magnitude of changes of 25(OH)D3 and 3-epi-25(OH)D3 in each studied group. For example, in the supplemented group, it was about 3 times increase in both vitamin D metabolites, and in the control, it was about 2 times increase. The outcome would be different if particular metabolites will respond differently (like in the 24,25(OH)2D3). Moreover, as it was introduced in the discussion (lines 326-330) the epi-25(OH)D3 may serve as a defense line of high levels of 25(OH)D3, which explain the similar direction of changes in both vitamin D metabolites regardless of the vitamin D concentration (supplementation vs control). Considering this comment, we used the opportunity to rewrite table 2. as the table presented statistics uncorrected to serum volume changes. Now the table fully represents the outcome showed in the figures. Accordingly, minor changes were introduced in the result section.

  1. Martineau, A. R., K. E. Thummel, Z. Wang, D. A. Jolliffe, B. J. Boucher, S. J. Griffin, N. G. Forouhi and G. A. Hitman (2019). "Differential Effects of Oral Boluses of Vitamin D2 vs Vitamin D3 on Vitamin D Metabolism: A Randomized Controlled Trial." J Clin Endocrinol Metab 104(12): 5831-5839.
  2. Freeman, J., K. Wilson, R. Spears, V. Shalhoub and P. Sibley (2015). "Performance evaluation of four 25-hydroxyvitamin D assays to measure 25-hydroxyvitamin D2." Clin Biochem 48(16-17): 1097-1104.

Minor points

Line 23: one dose of vitamin D"3".

Answer: thank you for this comment. We changed this to: “a single dose of vitamin D3

Line 113: Indicate your weight, please.

Answer: thank you for this point. The information about body mass was added to table 1.

Line 133: Please indicate the sunshine conditions on race day

Answer: thank you for this comment. The information about sunshine conditions (line 153: “Most of the time the sky was overcast.”, latitude (line 148: (latitude of 50° N)) and season (Line 146: “(19.07.2018)” was added.

Line 145: “ after "intake." should be removed.

Answer: done

Line 157: "SHIMADZU" LC-MS 8050.

Answer: thank you for this comment. The methods section has been corrected

Fig. 2: There are drastic increases 30-days after supplementation and these

peaks seems artificial. The reviewer recommends to plot from 0 to 30 days.

Answer: thank you for this point. Indeed the concentrations after 30-days increased drastically, but please notice that in the Pilot study, supplementation was performed twice (lines 137-138: “For this purpose, four physically active non-ultra-marathon runners (volunteers) took two doses of vitamin D3”(100,000 IU or 200,000 IU) 28 d apart). Firstly, on day 0 and secondly on day 28. Thus at 30thday there was drastic increase as some vitamin D metabolites did not make to return to baseline values and effect of second dose overlapped on first one. The same description is implemented in the figure’s description.

Fig. 3: There are three graphs where the "2" on the y-axis of the graphs are

not subscripted: 24,25(OH)2D3, 25(OH)D"2", and ratio 25(OH)D3:24,25(OH)2D3. 

Answer: sorry for this omission, corrected. Please notice that we also added the letter indication for each figure panel to increase the readability. In consequence, we changed the letter indication significance to common used symbols.

Reviewer 2 Report

Is this study a true repeated measures design? A repeated measures design is an experimental design where the same participants take part in each condition of the independent variable.

Could the title be more specific to reflect the results, rather than stating ‘changes’?

Abstract

It would be helpful if the results section of the abstract was supported with data for serum 25(OH)D3, 24,25(OH)2D3, and 3-epi-25(OH)D3 levels at least.

 The effect of exercise on serum vitamin D levels is possibly determined by the amount of stored vitamin D and degree of its mobilization. -  Is there any data to support this?

Introduction:

Overall, there is a presented rationale for the study with regards to vitamin D, but it would be helpful if the rationale for the effect of ultra-marathons were made clearer.

Lines 37-40: Is there any merit in mentioning the effect of low vitamin D status on physical performance at this point due to the nature of the investigation?

Methods

Line 102: ‘(not less than .’ is incomplete; how many races had the participants competed in?

Lines 102-4: ‘During all testing periods and 1 week before testing, the participants refrained from alcohol, caffeine, guarana, theine, tea, chocolate and any other substance intake, that may potentially influence exercise performance’ -  Was this assessed?  What was the rationale for seven days, and did this reflect usual practise?

Report age as an integer, and can height be measured accurately to two decimal places?

It would be useful to state the time of year that the race took place due to the potential seasonal variation in vitamin d status.

It might be helpful to conduct an ANCOVA to control for baseline vitamin D status

Results

Figure 3 looks much clearer than figure 2. Could the authors consider presenting figure 2 in a similar manner to figure 3?

The data in table 2 could be incorporated into the text.

Discussion

The discussion provides a nice overview of the results, but there needs to be some development of the mechanistic underpinnings and practical implications.

I hope that the authors find the above comments helpful and in the constructive manner they are intended.

Author Response

R2

Is this study a true repeated measures design? A repeated measures design is an experimental design where the same participants take part in each condition of the independent variable.

Could the title be more specific to reflect the results, rather than stating

‘changes’?

Answer: thank you for this point. Considering the study design, we used repeated measures design by Two-way ANOVA of repeated measures which is the most common approach to assess differences in an outcome serially measured at different time points in the same subjects (Sullivan 2008, Schober and Vetter 2018). We could use the experimental repeated measures design by using the same participants (crossover study design), however, it would be difficult to control the investigation on the same participants in two ultra-marathon runs as the conditions including distance, weather, seasons and etc. would vary significantly.

We have to change the title to reflect the results: Ultra Marathon-induced Increase in Serum Levels of Vitamin D Metabolites: A Double-blind Randomised Controlled Trial

  1. Schober P, Vetter TR. Repeated Measures Designs and Analysis of Longitudinal Data: If at First You Do Not Succeed-Try, Try Again. Anesthesia and analgesia (2018) 127(2):569-75.
  2. Sullivan LM. Repeated measures. Circulation (2008) 117(9):1238-43.

Abstract

It would be helpful if the results section of the abstract was supported with data for serum 25(OH)D3, 24,25(OH)2D3, and 3-epi-25(OH)D3 levels at least.

 Answer: thank you for this comment. We supplemented the abstract with the following data in the result section  (lines 27-29): In both groups of runners, serum 25(OH)D3, 24,25(OH)2D3, and 3-epi-25(OH)D3 levels significantly increased by 30%, 63% and 182% after the ultramarathon, respectively.

The effect of exercise on serum vitamin D levels is possibly determined by the amount of stored vitamin D and degree of its mobilization. -  Is there any data to support this?

 Answer: thank you for this comment. Reviewer is right, while we did not used methods that allow for direct quantification of vitamin D storage, we deleted the sentence.

Introduction:

Overall, there is a presented rationale for the study with regards to vitamin D, but it would be helpful if the rationale for the effect of ultra-marathons were made clearer.

Answer: thank you for this comment. To supplement the rationale for the ultra-marathon we added in lines 81-88: “Till now, knowledge about vitamin D metabolism associated with strenuous exercise like ultra-marathon is limited (Kasprowicz, Ratkowski et al. 2020). Running and walking for extreme durations, so-called ultra-marathons, have become increasingly popular in the last years throughout the world, particularly in the USA, Europe, Japan and South Africa  (Hoffman et al. 2010). Hence, it seems important to evaluate the effect of this type of prolonged exercise on the physiological responses of the human body, both under conditions of supplementation and when is not provided. Especially that vitamin D supplementation could prevent exercises induced inflammation process and other adverse body reactions”.

Moreover, we added a reference to the existing sentence (lines 88-90): “We proposed that ultra-marathon, which alters the production of hundreds of exercises (Knechtle and Nikolaidis 2018)and has the potential to reduce the amount of adipose tissue (Tiller, Roberts et al. 2019), influences vitamin D metabolism”.

  1. Hoffman, M.D.; Ong, J.C.; Wang, G. (2010). “Historical analysis of participation in 161 km ultramarathons in North America”. The International journal of the history of sport, 27, 1877-1891.
  2. Kasprowicz, K., et al. (2020). "The Effect of Vitamin D3 Supplementation on Hepcidin, Iron, and IL-6 Responses after a 100 km Ultra-Marathon." Int J Environ Res Public Health 17(8).
  3. Knechtle, B. and P. T. Nikolaidis (2018). "Physiology and Pathophysiology in Ultra-Marathon Running." Front Physiol 9: 634.
  4. Tiller, N. B., J. D. et al. (2019). "International Society of Sports Nutrition Position Stand: nutritional considerations for single-stage ultra-marathon training and racing." J Int Soc Sports Nutr 16(1): 50.

Lines 37-40: Is there any merit in mentioning the effect of low vitamin D status on physical performance at this point due to the nature of the investigation?

Answer: thank you for great point. We added sentence considering vitamin D deficiency on performance, line 43-44: “Moreover, also more and more athletes show the low vitamin D status, which may negatively impact the health, performance and training efficiency of athletes”.

Methods

Line 102: ‘(not less than .’ is incomplete; how many races had the participants competed in?

Answer: sorry for this omission, we completed the sentence, lines 113-114: “All runners had previous ultra-marathon race experience (not less than two) “

Lines 102-4: ‘During all testing periods and 1 week before testing, the participants refrained from alcohol, caffeine, guarana, theine, tea, chocolate and any other substance intake, that may potentially influence exercise performance’ -  Was this assessed?  What was the rationale for seven days, and did this reflect usual practise?

Answer: the authors made participants aware of the influence of mentioned substance on the study and relied on their declaration. Additionally mentioned substances may have visible influence on sport performance and may be associated to the type of physiological response even during a training period. Most researchers (Del Coso et all 2008;  Del Coso, J. et all 2012) in their research protocol use one or two days period when the participants’ refrained from alcohol, caffeine, and any other stimulant substances. But in our situation, a 7-day break period when the participants’ refrained from stimulants was dictated by the fact that most of the professional or semi-professional runners during the week preceding the sports competition, no longer does intense training sessions and performing only short, medium, or low insensitivity training units, trying to regenerate as much as possible in the period before the start, and cooling down additional stimulations.  Furthermore, due to the fact that for example in North America, 80% to 90% of all adults use caffeine regularly (EFSA Panel on Dietetic Products, 2015), we wanted to eliminate some of the problematic symptoms of most popular caffeine withdrawal syndrome (such as headache, fatigue, decreased energy and activeness, and many more). Those symptoms are mostly observed during the first 12-24h after abstinence of caffeine and should only last between two to nine days (Juliano & Griffiths, 2004). We shortened the 9 days period only to 7 days (as a weekly period of low and medium intensity training units) in the whole population to refrain from alcohol, caffeine, guarana, theine, tea, chocolate, and any other substance intake due to the fact that whole population in the preliminary interview reported very little consumption of potentially stimulating substances (e.g. caffeine, its derivatives, precursors and others) and did not have any symptoms of a caffeine withdrawal syndrome in firs 72h.

1.     Del Coso, J., Muñoz-Fernández, V. E., Muñoz, G., Fernández-Elías, V. E., Ortega, J. F., Hamouti, N., Barbero, J. C., & Muñoz-Guerra, J. (2012). Effects of a caffeine-containing energy drink on simulated soccer performance. PloS one, 7(2), e31380. 2.     Del Coso, J., Estevez, E., & Mora-Rodriguez, R. (2008). Caffeine effects on short-term performance during prolonged exercise in the heat. Medicine and science in sports and exercise, 40(4), 744.3.     Juliano, L. M., & Griffiths, R. R. (2004). A critical review of caffeine withdrawal: empirical validation of symptoms and signs, incidence, severity, and associated features. Psychopharmacology, 176(1), 1-29.4.     EFSA Panel on Dietetic Products, Nutrition and Allergies (NDA). (2015). Scientific Opinion on the safety of caffeine. EFSA Journal, 13(5), 4102.

Report age as an integer, and can height be measured accurately to two decimal places?

Answer: thank you for this comment. We change presentation of age accordingly. The body height was measured as integer (cm), however values presented in table refer to mean and standard deviations, thus the occurrence of values in two decimal places.

It would be useful to state the time of year that the race took place due to the potential seasonal variation in vitamin d status.

Answer: thank you good point. The information about conditions that may impact the Vitamin D production was added in line 153: “Most of the time the sky was overcast.”, line 148: (latitude of 50° N) Line 146: (19.07.2018)”  was added.

It might be helpful to conduct an ANCOVA to control for baseline vitamin D status

Answer: thank you for this comment. We agree that the ANCOVA approach would be beneficial for controlling the baseline values, however our Statistical software package was unable to perform Two-way ANCOVA testing, only the one-way ANCOVA. In this manner, we chose the two-way ANOVA approach, as the outcome is more informative for the study aim. Moreover, our data have shown that there was no significant difference in baseline values between groups, thus the impact of baseline vitamin D status should be limited.

Results

Figure 3 looks much clearer than figure 2. Could the authors consider

presenting figure 2 in a similar manner to figure 3?

Answer: thank you for this point. Figures 1 and 2 were redone to meet the standard of Figure 3.

The data in table 2 could be incorporated into the text.

Answer: thank you for this point. We used the table 2 and figure 3. to reduce the text in result section and show all effects in more clear way. However, due to comment we, decided to supplement the text with information on the main effects of supplementation (lines 232-236): “Two-way ANOVA revealed also a significant group effect in 25(OH)D3, 3-epi-25(OH)D3 and 25(OH)D3 : 24,25(OH)2D3 ratio (Table 2). The UM-S group showed: 30.1% higher concentration of 25(OH)D3, 61.8% higher concentration of 3-epi-25(OH)D3 and 24.7% higher ratio of 25(OH)D3 to 24,25(OH)2D3 in comparison with UM-C group.”

Discussion

The discussion provides a nice overview of the results, but there needs to be some development of the mechanistic underpinnings and practical implications.

Answer: Thank you for the good point. Some changes has been introduced in discussion section, please look at lines 331- 338:

“The consumption of vitamin during ultramarathon is a common nutritional habit and along with the growing interest in this type of physical activity, has been studied (Nikolaidis et al. 2018; Knez and Peake 2010) It should be highlighted that intake typical antioxidants like vitamin C and E, as anti-inflammatory and antioxidative factors in endurance training could even blunt training adaptations and attenuate some of the cellular adaptations in skeletal muscle (Costaet al. 2019). Furthermore those vitamins consumption did not affect physiological aspects related to sport performance and did not improve sport results (Taghiyar et al. 2013). That’s why it is a need for searching proper supplementation methods, where vitamin D supplementation can be beneficial for ultramarathon runners health.”

 …and lines 343-348: “All of these data indicate that these changes in vitamin D metabolism are a physiological response to endurance exercise. The changes are affected by vitamin D status thus one can predict that low adipose or skeletal muscle vitamin stores may negatively influence physiological response to exercise. However more work is needed to explore the role of vitamin D metabolites in physiological response to exercise.”  

  1. Nikolaidis, P. T., Veniamakis, E., Rosemann, T., & Knechtle, B. (2018). Nutrition in Ultra-Endurance: State of the Art. Nutrients, 10(12), 1995.
  2. Knez, W. L., & Peake, J. M. (2010). The prevalence of vitamin supplementation in ultraendurance triathletes. International journal of sport nutrition and exercise metabolism, 20(6), 507-514.
  3. Costa, R. J., Knechtle, B., Tarnopolsky, M., & Hoffman, M. D. (2019). Nutrition for ultramarathon running: Trail, track, and road. International Journal of Sport Nutrition and Exercise Metabolism, 29(2), 130-140.
  4. Taghiyar, M.; Ghiasvand, R.; Askari, G.; Feizi, A.; Hariri, M.; Mashhadi, N.S.; Darvishi, L. (2013) The effect of vitamins C and e supplementation on muscle damage, performance, and body composition in athlete women: a clinical trial. International journal of preventive medicine, 4, S24-30.

I hope that the authors find the above comments helpful and in the constructive manner they are intended.

Answer: thank you very much for all comments. They help increase quality of the manuscript. We have hope that we  answered your questions and comments satisfactorily.

Round 2

Reviewer 2 Report

I believe that the authors have addressed my previous comments and should be applauded for this nice applied investigation.